# Effect of On-Duty Resistance Training Fatigue on Neuromuscular Function in Structural Firefighters

**DOI:** 10.3390/healthcare13111278

**Published:** 2025-05-28

**Authors:** Jamal L. Thruston, Stuart A. Best, Nicholas R. Heebner, Lance M. Bollinger, Mark G. Abel

**Affiliations:** 1First Responder Research Laboratory, University of Kentucky, Lexington, KY 40506, USA; 2Sports Medicine Research Institute, University of Kentucky, Lexington, KY 40509, USA; nick.heebner@uky.edu; 3Department of Kinesiology and Health Promotion, University of Kentucky, Lexington, KY 40506, USA; stuart.best@uky.edu (S.A.B.); lance.bollinger@uky.edu (L.M.B.)

**Keywords:** postural control, postural sway, heavy resistance training, circuit training, functional balance test, time to stabilization, single leg drop landing, isometric midthigh pull

## Abstract

Background: Participation in on-duty exercise is critical to enhance firefighter safety and readiness. However, these sessions are often interrupted with emergency responses and require firefighters to work in a fatigued state that may increase injury risk. Objective: To assess the impact of on-duty resistance training on neuromuscular function. Methods: A sample of 18 firefighters (Age: 38.8 ± 8.0 y; Body fat: 24.9 ± 7.0%) completed three testing sessions, separated by at least 72 h to compare the effects of circuit (CT) versus heavy resistance training (HRT) fatigue on neuromuscular function. During Session 1, anthropometrics and familiarization trials of balance and neuromuscular function were completed, which included single-leg drop landing (SLDL), postural sway (PS), and modified Functional Balance Test (mFBT). Sessions 2 and 3 were randomized, where participants completed either HRT or CT. Isometric midthigh pull (IMTP), long jump (LJ), and lower body power (LBP) tests were conducted pre- and immediately post exercise, whereas static and dynamic balance assessments were conducted pre- and 10 min post exercise to simulate an emergency response time course. Repeated measures ANOVA, effect sizes, and difference scores were used to analyze the effects of condition and time. The level of significance was set at *p* < 0.05. Results: CT decreased IMTP, LJ, and LBP, whereas HRT decreased LJ and LBP (*p* ≤ 0.001, ES ≥ 0.476). Despite several significant condition by time interaction effects on balance outcomes, there were no differences within CT or HRT over time (*p* ≥ 0.066). Conclusions: These findings suggest that on-duty resistance training reduces firefighters’ power and/or strength immediately post exercise but does not influence most firefighters’ balance 10 min post exercise. Thus, firefighters are recommended to perform resistance training on-duty during low emergency call volume times.

## 1. Introduction

Structural firefighting requires the completion of rigorous occupational tasks in austere environments that increase injury risk. Injuries are a substantial problem in the fire service as about 65,000 injuries are reported annually in the United States [1]. Specifically, regarding causes of fireground injuries, 22% are the result of jumps, slips, trips, and falls, and 31% are related to overexertion or strain. Regarding non-fireground injuries, 57% are from strains, sprains, and muscular pain, which are the leading nature of injury [1]. Fiscally, firefighter injuries cost the United States USD 1.6 to 5.9 billion annually and approximately USD 50,000 to USD 200,000 per fire department per year [2]. These costs are associated with workers’ compensation claims and expenses related to backfilling the injured firefighter’s position. Thus, injury prevention and safety programs are needed in the fire service.

To complete occupational tasks safely and effectively, the National Fire Protection Association (NFPA) recommends regular participation in exercise while on-duty. In support of this recommendation, empirical data have demonstrated that on-duty exercise interventions enhance occupational physical ability and physical fitness. Specifically, Pawlak et al. (2015) demonstrated that on-duty exercise can improve performance in a simulated fire ground test, and yield improvements in body composition and body mass index (BMI) [3]. Despite the evidence to support the incorporation of on-duty exercise, it is important to consider the acute effects of residual fatigue if an exercise session is interrupted by an emergency response, as this has been shown to decrease occupational performance [4,5] and may increase injury risk. For instance, Mason et al. (2023) reported a 45% decrease in work rate following high-intensity functional training in a sample of 7 male firefighters, whereas Dennison et al. (2012) found a 10% decrement in work rate after a circuit training bout in 12 fit firefighters. Interestingly, Dennison and colleagues [5] noted that these fit but fatigued firefighters were still able to perform occupational tasks at a faster work rate than most unfit, non-fatigued firefighters, demonstrating the importance of regular participation in on-duty exercise to develop occupationally relevant fatigue tolerance.

Despite the aforementioned work evaluating the impact of exercise-induced fatigue on occupational performance, there is limited research evaluating the impact of resistance training fatigue on neuromuscular function, as fatigue is considered a potential risk factor for slip, trip, and fall injuries [6]. In one such study, Trivisonno and colleagues (2021) [7] reported reduced muscular strength and power metrics among firefighters 24–72 h following a bout of on-duty circuit training. However, to the best of our knowledge, there is a lack of research directly examining the impact of resistance training on balance and neuromuscular function immediately following exercise among firefighters. Research in other populations indicates that exercise-induced neuromuscular fatigue can negatively impact postural control [8] and that static and dynamic postural control assessments are related to musculoskeletal injury risk [9,10,11]. Thus, utilizing postural control assessments provides valuable insight regarding injury risk.

Furthermore, it is important to discern if certain types of on-duty resistance training may be contraindicated to offer evidence-based recommendations to the fire service. Specifically, two commonly utilized training modalities, circuit and heavy resistance training, are recommended to develop occupational readiness among structural firefighters [12]. However, circuit training yields greater levels of physiological (heart rate, oxygen consumption, and blood lactate) and perceptual strain compared to traditional resistance training [13], and these factors are associated with degradations in balance [14]. Thus, circuit training may have a greater impact on firefighters’ balance and neuromuscular function. Therefore, the primary aim of this study was to assess the effect of circuit and heavy resistance training induced fatigue on firefighters’ functional balance and neuromuscular function. We hypothesized that circuit training, but not heavy resistance training, would decrease functional balance and neuromuscular function in firefighters due to its higher fatigue potential. We also hypothesized that muscular strength would correlate to decrements in functional balance and neuromuscular function. Regarding functional balance, the Functional Balance Test (FBT) has been developed and utilized in several studies to assess firefighters’ balance in a more occupationally relevant manner [15]. However, there are limited published data evaluating the validity and reliability of this assessment in firefighters. Therefore, a secondary aim of this study was to evaluate the face and criterion-related validity and test–retest reliability of a modified FBT (mFBT). Based on pilot data, we hypothesized that the mFBT would yield acceptable levels of validity and reliability.

## 2. Materials and Methods

### 2.1. Research Design

This study employed a crossover design to assess the acute impact of two resistance training stimuli on neuromuscular outcomes in structural firefighters. The independent variables included heavy resistance and circuit training modalities. The dependent variables included muscular strength and power, mFBT, postural sway, and single-leg time to stabilization. Static and dynamic balance force plate assessments were chosen due to their sensitivity, ease of use, and association with musculoskeletal injuries [9,10,11]. Strength and power assessments were chosen for ease of use, reliability, and association with injuries in tactical populations [16,17]. The mFBT was utilized due its relevance to movement patterns performed on a fireground.

### 2.2. Participants

A convenience sample of 19 career structural firefighters were recruited to participate in this study from July 2023 to April 2024. One participant voluntarily withdrew from this study due to unrelated discomfort from a previous injury. Thus, 18 male firefighters (Age: 38.8 ± 8.0 y; Height: 179.4 ± 5.4 cm; BMI: 32.1 ± 5.1 kg/m^2^, Body fat: 24.9 ± 7.0%) completed this study. Inclusion criteria to participate were as follows: (1) full-time employed firefighter; (2) medically cleared for active duty; and (3) between 18 and 60 y of age. Participants were excluded from this study if (1) they presented with a current or previous musculoskeletal injury within the last 3 months; (2) had respiratory, metabolic, or neuromuscular disorders that would impact occupational performance or contraindicate exercise participation. Participants completed a Physical Activity Readiness Questionnaire (PAR-Q) to determine readiness to safely participate in physical activity and exercise testing. Institutional Review Board approval (Protocol #84866, Approval: 18 April 2023) was obtained prior to study initiation. Participants provided written informed consent prior to participation in this study.

### 2.3. Procedures

Participants completed three testing sessions separated by at least 72 h at a university laboratory. The sessions were typically conducted while on-duty. Table 1 provides a summary of the testing session composition and protocol sequence. Participants were asked to refrain from exercise for at least 2 days prior to all testing sessions. During Session #1, readiness questionnaires, anthropometric measurements, and familiarization trials of neuromuscular balance and strength and power assessments were completed. Finally, participants performed 5 repetition maximum (RM) protocols for each exercise (bench press, deadlift, bent over row, squat, and shoulder press) to be included in the circuit and heavy resistance training protocols.

During Sessions 2 and 3, participants completed either a standardized circuit or heavy resistance training protocol. The order of the resistance training protocols was block randomized. The isometric midthigh pull (IMTP), standing long jump, postural sway, single-leg landing, and mFBT assessments were performed prior to the resistance training protocol. Immediately following the resistance training protocol, participants performed the IMTP and standing long jump to quantify the impact of each protocol on maximal isometric strength and power, respectively. Firefighters were then provided with a 10 min recovery period meant to simulate a typical emergency response duration, where firefighters donned department issued personal protective equipment (PPE, i.e., boots, turnout gear, helmet, gloves, and self-contained breathing apparatus (SCBA)). Following 10 min, postural sway, single-leg landing, and the mFBT were assessed. Before, during, and after exercise, rating of perceived exertion (RPE) and heart rate were assessed. Blood lactate was assessed before and 5 min post-exercise.

### 2.4. Anthropometric Measurements

Standing height and body mass were measured without shoes and in physical training attire. Height was measured with a stadiometer (Detecto 3P7044, Webb City, MO, USA) to the nearest 0.5 cm. Body mass was measured using a digital scale to the nearest 0.1 kg (Medline Digital Step-On Scale, Northfield, IL, USA). Body composition was measured using a tetrapolar dual-frequency bioelectrical impedance analyzer (BIA; BodyStat 1500, BodyStat Ltd., Isle of Man, UK). Specifically, electrodes were placed on the participants’ right wrist, hand, ankle, and foot. The participant’s sex, age, body mass, height, waist circumference, hip circumference, and activity level were entered into the device to provide the manufacturer’s proprietary prediction equation of relative body fat. Waist and hip circumferences were measured with a non-elastic tape measure at the narrowest part of the torso between the umbilicus and xiphoid process and at the widest part of the torso around the buttocks, respectively [18]. Circumference measures were taken in duplicate and the mean value was used for analysis.

### 2.5. Swedish Occupational Fatigue Inventory

The Swedish Occupational Fatigue Inventory (SOFI) was utilized in Sessions #2 and #3 to capture of the firefighters’ perceived lack of energy, physical exertion, physical discomfort, lack of motivation, and sleepiness to compare between conditions. Internal consistency of the SOFI has been reported to be moderate to high (r = 0.68–0.93) [19].

### 2.6. Hydration

Hypohydration has been reported to impact postural sway outcomes [14], thus urine specific gravity was measured using a digital refractometer (PAL10S, Atago, Tokyo, Japan). Participants were assessed at the beginning of Sessions #2 and #3. This refractometer has been reported to yield valid (r = 0.97, *p* < 0.001) and reliable measures [20]. Euhydration was considered ≤ 1.020 as recommended by the American College of Sports Medicine [21].

### 2.7. Postural Sway

Postural control outcomes (mean velocity total, anterior–posterior, and medial-lateral; total excursion; anterior–posterior and medial-lateral center of pressure (COP) range) were assessed using (1) 30 s trial with double leg stance (eyes open and closed) and (1) 15 s trial using single leg (eyes open) on force plates (Vald, Newstead, Australia), which has demonstrated excellent reliability (ICC = 0.90–0.98) [22]. There is also excellent test–retest reliability within non-fatigued (r = 0.82) and post-exercise fatigued states for the aforementioned outcomes (r = 0.87) [23]. Specifically, participants donned PPE and completed double-leg stance trials with eyes open and closed and single-leg stance (left and right) with eyes open. Force plate sampling rate was set at a frequency of 1000 Hz. Leg dominance was identified and defined as the leg used to kick a ball. To increase reliability, a marking of a black cross was placed on a wall across from the participant at eye level for the eyes-open condition. Participants were asked to maintain hands on hips during all single-leg conditions, whereas arms were extended in a neutral position at the sides during bipedal conditions. Feet were parallel, and during single-leg stance, the knee was flexed to 90 degrees with sole of foot positioned vertically. Trials were considered incomplete and repeated if participants could not maintain balance, hands came off hips, both feet touched the ground, legs contacted each other, and/or if a hop was added to support stability.

### 2.8. Single-Leg Drop Landing

Functional single-leg stability was measured by the single-leg landing assessment as recommended in previous research [24]. Participants aligned their toes with the edge of a box (Height: 30 cm) and stepped off the box onto a force plate with the assessed leg. Upon landing, the rear knee was flexed with the sole of the foot positioned vertically. Participants were required to maintain a single leg stance for 2–3 s. Trials were considered incomplete and repeated if the participants could not maintain balance for 2–3 s, if hands came off the hips, if both feet touched the ground, and/or if a hop was added to maintain the stance. Two repetitions were completed per leg, and time to stabilization was recorded. This protocol yields acceptable intra-session (intratester) reliability (ICC = 0.72) and excellent inter-day (intratester) reliability (ICC = 0.83) [24]. Force plate sampling rate was set at a frequency of 1000 Hz.

### 2.9. Modified Functional Balance Test

The mFBT is an occupationally relevant dynamic balance assessment that has been utilized in different configurations with firefighters [15,25]. Based on pilot data and feedback from career structural firefighters in the present study, the mFBT was further modified to enhance the test’s relevancy for firefighters, by incorporating a raised obstacle to step over, which simulates stepping over objects on the fireground and while carrying a sledgehammer (Mass = 4.1 kg; Trusty-Cook, Indianapolis, IN, USA) to replicate carrying an asymmetric load. A figure of the mFBT is displayed in Figure 1. The face validity of the mFBT was assessed after Session #3 by identifying the mFBT as “not relevant”, “somewhat relevant”, or “relevant”. Nine trials of the mFBT were performed in PPE to facilitate familiarization, assess test–retest reliability, and obtain stable values. As such, the mean Performance Index value from trials 4–6 of each condition were used for data analysis.

Participants were instructed to complete each mFBT trial as quickly as possible without making errors (i.e., touching an obstacle, feet coming off obstacle course, etc.). To complete the mFBT, participants began by standing on top of a raised platform (Length × width × height: 60 cm × 60 cm × 10 cm) in full PPE with their toes aligned with the leading edge of the platform. The trial time began upon the participant’s first movement and was measured manually using a stopwatch (Model 495, Ultrak, Gardena, CA, USA). The participants then ambulated across a 3.65 m length plank (Height × width: 5 cm × 15 cm) and stepped over an obstacle set at 20% of the participant’s standing height and is composed of 2 horizontal bars placed 35 cm apart. Participants then maneuvered under an obstacle set at 75% of standing height and walked to the second platform (Length × width × height: 60 cm × 60 cm × 15 cm). The participant stepped onto the second platform, turnaround within a 45 cm × 45 cm border positioned on the leading edge of the platform and returned through the obstacles to the first platform. The time trial was completed once the participants turned around and aligned the toes with the leading edge of the platform.

A one second penalty was added to the total mFBT time for each minor error, whereas a two second penalty was added for each major error. Minor errors included stepping on the platform borders, touching a hand or foot on the ground to maintain balance on the plank or platforms, or touching an obstacle without displacing the bar. Major errors include knocking a bar off the barrier or completely falling off the plank or platform. The aggregate time, including time-error penalties for the mFBT is referred to as the Performance Index. Digital cameras were used to record trials and placed near the end of the platforms to identify errors during post-session video analysis. After completing 6 trials, participants completed an additional 3 trials with a cognitive load simulation task.

### 2.10. Cognitive Load Simulation Task

An occupationally relevant cognitive load was integrated during pre- and post-resistance training protocols while performing the mFBT to simulate firefighters’ need to perform balance tasks while consuming operational information. Specifically, an audio file played through a portable speaker (JBL Flip 6, HARMAN, Stamford, CT, USA) that mimicked an emergency radio broadcast during the final 3 trials of the mFBT (i.e., Trials #6–9). The audio communication provided descriptions of a structure fire and relevant details that were randomized for each trial. Specifically, the script was as follows: “Engine 92 is on scene. [Building Address Number] [Street Name] is a 20-story ordinary construction commercial building with a flat roof. There is heavy smoke coming from the [Direction] side of the [Floor Number] floor, and a visible fire from the [Direction] side of the [Floor Number] floor.” Immediately following the mFBT the participant was asked to recall 6 randomized details (e.g., structure address number and street name, direction of smoke and fire, and floor numbers of smoke and fire). The number of correct responses and total variables were recorded and expressed as a percentage:(Correct responses/Total responses) × 100.

### 2.11. Isometric Midthigh Pull

Muscular performance and fatigue status was assessed with an IMTP using a flywheel resistance training device (Exxentric kBox4 Pro, Bromma, Sweden). A t-bar (Mass: 0.60 kg) was tethered to the platform with two force plates positioned on top of the platform. The force plates (Vald, Newstead, Australia) measured absolute and relative peak force (N) at a sampling frequency of 1000 Hz. The test–retest reliability of IMTP absolute peak force ranges from 0.84 to 0.99 (median ICC = 0.97) [26]. To perform the assessment participants started in a partial squat position with hips flexed to 145° [27], and knees flexed to 145°, measured by a goniometer. The t-bar was grasped with a closed pronated grip with the arms extended such that the bar was positioned at mid-thigh. Participants completed 2 warm-up repetitions of 50% and 75% effort, respectively. Then, participants completed 3 trials utilizing maximal effort. The greatest value was used in data analysis.

### 2.12. Standing Long Jump

A standing long jump was performed to estimate lower body power. The inter-session test–retest reliability of the long jump in this sample was ICC = 0.93. Specifically, firefighters were instructed to jump as far forward as possible, and the distance traveled was measured from the rearmost heel. Trials were considered invalid if the participant slid upon landing, lost balance, or took an extra step. The trial with greatest distance of three correctly performed trials was included in the analyses. Lower body power was calculated using the following prediction equation:Power (W) = [(32.49 × jump length (cm)) + (39.69 × body weight (kg))] − 7608.

This equation has demonstrated acceptable levels of validity [28].

### 2.13. Strength Testing

Prior to strength testing, participants performed a general and dynamic warm-up led by the principal investigator. Participants then performed 5 RM assessments on the following exercises: barbell bench press, trap bar deadlift, barbell bent-over row, smith machine back squat, and seated dumbbell overhead press. Due to lower back stiffness, 1 participant chose not to compete a barbell bentover row and instead was allowed to complete a barbell bench pull.

To complete the 5 RM protocol for each exercise, participants performed 10 repetitions with a light load. After a rest period of 1 min, 5 repetitions were performed using a moderate load. After a rest period of 1 min, 5 repetitions were performed with a near-maximal load. Following a 2 min recovery period, participants perform a 5 RM set. If additional or fewer repetitions were completed the load was modified for each subsequent set until a 5 RM was achieved. Two minutes of recovery were provided prior to each 5 RM set. The 5 RM load was applied to a prediction table [29] based off Mayhew et al.’s [30] 1-RM formula to estimate the 10 RM load. The Mayhew et al. [30] 1 RM formula is as follows:1 RM = Repetition Weight/[[52.2 + 41.9e^(−0.055(# Reps))]^/100].

The formula has been cross validated in college men (r = 0.96, *p* < 0.01) to estimate the 1-RM [30].

### 2.14. Resistance Training Protocols

Participants completed heavy resistance training and circuit training sessions in a randomized order and on separate days separated by at least 72 h. For both exercise sessions, participants performed barbell bench press, trap bar deadlift, barbell bent over row, smith machine back squat, and dumbbell overhead press. These exercises were selected based on their functional nature and specificity for fireground tasks. The heavy resistance training protocol consisted of 3 sets of 5 repetitions with 100% of the ascertained 5 RM load and 2 min inter-set and inter-exercise recovery periods. Exercises were completed in sequential order. For the circuit training protocol, 2 circuits composed of 1 set of 10 repetitions with 90% of the estimated 10 RM load of all exercises were completed with 45 s of inter-exercise recovery. On average, the volume of the circuit training session was 9.6% greater than the heavy resistance training session.

### 2.15. Perceptual and Physiological Measures

A global rating of perceived exertion (RPE) was assessed before and immediately after each resistance training protocol using a 0–10 category-ratio scale. This scale has yielded high levels of reliability (r = 0.95) [31]. Mean and peak session heart rate were monitored continuously during the resistance training protocols with a heart rate monitor (Polar H10, Polar Electro, Kempele, Finland) which was placed around the participant’s chest. Heart rate data were transmitted wirelessly to an electronic device, where data were recorded. Blood lactate was measured prior to and 5 min post resistance training using a blood lactate analyzer (LactatePlus, Nova Biomedical, Waltham, MA, USA). This device has yielded valid (r > 0.99) and reliable (r ≥ 0.99) blood lactate measurements [32]. Specifically, universal precautions were used and a blood sample was taken from the fingertip. The first drop of blood was discarded, and the second drop was applied to an assay strip. The accuracy of the blood lactate analyzer was checked using low (1.0–1.6 mmol/dL) and high (4.0–5.4 mmol/dL) control solutions prior to each testing session.

### 2.16. Statistical Analysis

Basic statistics were used to describe measures of central tendency (mean and median) and dispersion (standard deviation and range) for dependent and descriptive variables. To assess the first aim, two-way (condition × time) repeated measures ANOVA were used to assess differences in dependent variables within and between resistance training conditions. Post hoc analyses were conducted using paired sample t-tests. The statistical assumptions of normality and sphericity were analyzed using the Shapiro–Wilk Test and Mauchly’s Test, respectively. If the sphericity assumption was not met, the Greenhouse Geisser correction was applied. Effect sizes were assessed using partial eta squared (η_p_^2^) and interpreted as small effect: 0.01–0.05; medium effect: 0.06–0.13; large effect: ≥0.14. The level of significance was set at *p* < 0.05. Relative difference scores were used to describe within group changes in pre- versus post-resistance training outcomes relative to the pre-resistance training outcome, using the following formula:% difference = ([post-trial outcome − pre-trial outcome]/pre-trial outcome) × 100%.

A Wilcoxon signed-rank test was used to compare SOFI outcomes prior to the circuit and heavy resistance training sessions. Minimal difference (MD) analysis was used to identify clinically relevant changes, on an individual level, for all dependent variables. Regression analysis of pre-exercise dependent variables from Sessions #2 and 3 were used to identify the standard error of the measurement (SEM) from the mean squared error. Then, the MD threshold was calculated using the following formula [33]:MD = SEM × 1.96 × √2.

Post-exercise neuromuscular function and balance outcomes were then compared to the MD threshold. Pearson product moment correlation coefficients were used to identify significant fitness and anthropometric correlates of the decrement in neuromuscular outcomes. If assumptions of normality were not met, Spearman’s rank correlation coefficients (r_sp_) were used.

To assess the secondary aim, criterion related validity was assessed by evaluating Pearson Product Moment correlations between the mean mFBT’s Performance Index value from trials 4–6 versus force plate-based postural sway and time-to-stabilization outcomes. Face validity of the mFBT was assessed based on the proportion of participants’ rating the mFBT to be “not relevant”, “somewhat relevant”, or “relevant”. Inter-session and intra-session test–retest reliability of the mFBT’s Performance Index were assessed using intraclass correlation coefficients (ICC_2,1_). An apriori sample size estimate was conducted based on pilot data that suggested 20 participants were needed given a medium effect (Cohen’s d = 0.67) of resistance training on postural sway outcomes, with power set at 0.8 and alpha at 0.05. Statistical analyses were performed using the Statistical Package for Social Sciences (SPSS, Version 29).

## 3. Results

Prior to exercise, there were no differences between circuit versus heavy resistance training sessions in SOFI outcomes (Table 2). The physiological and perceptual outcomes of circuit and heavy resistance training sessions are provided in Table 3. Following exercise, circuit training produced greater blood lactate (*p* < 0.001), peak (*p* = 0.002) and average heart rate (*p* < 0.001), dyspnea (*p* = 0.005), and RPE (*p* = 0.007) compared to heavy resistance training.

The primary aim evaluated the impact of circuit versus heavy resistance training on neuromuscular outcomes. Table 4 presents the relative difference values and comparison by training condition for all dependent variables. There was a significant main effect of time for peak force (F(1,17) = 15.450, *p* = 0.001, η_p_^2^ = 0.476, Power = 0.959), long jump (F(1,17) = 20.373, *p* < 0.001, η_p_^2^ = 0.545, Power = 0.989) and lower body power (F(1,17) = 20.373, *p* < 0.001, η_p_^2^ = 0.545, Power = 0.989). Post hoc analysis indicated that circuit training decreased peak force, long jump distance, and lower body power (*p* ≤ 0.001), whereas heavy resistance training decreased long jump distance and lower body power (*p* < 0.001), but not peak force (*p* = 0.102).

There were conditions by time interaction effects for peak force (F(1,17) = 8.756, *p* = 0.009), long jump distance (F(1,17) = 7.074, *p* = 0.017), and lower body power (F(1,17) = 7.074, *p* = 0.017) demonstrating greater decrements in circuit training from pre- to post-exercise compared to heavy resistance training. In addition, there were significant interaction effects for postural sway—eyes open overall mean velocity (F(1,17) = 4.759, *p* = 0.043), anterior–posterior mean velocity (F(1,17) = 4.473, *p* = 0.049) and total excursion (F(1,17) = 4.641, *p* = 0.046), as well as postural sway—eyes closed medial-lateral mean velocity (F(1,17) = 5.450, *p* = 0.032) and medial–lateral COP range (F(1,17) = 8.066, *p* = 0.011) indicating greater decrements in circuit than heavy resistance training from pre- to post exercise. However, post hoc analysis indicated that these post-circuit training postural sway outcomes were not different from baseline (*p* = 0.066–0.098).

### 3.1. Minimal Difference (MD)

Table 5 presents the percentage of participants who exceeded the MD threshold. For decrements in peak force, 22% of circuit training participants exceeded the lower MD threshold, whereas 6% exceeded the lower MD threshold in the heavy resistance training condition. Regarding increases in peak force, 0% exceeded the upper MD threshold in circuit training, whereas 6% exceeded the MD in heavy resistance training. Regarding decrements in power metrics, 50% of circuit training participants exceeded the lower MD threshold, whereas 0% exceeded the lower MD threshold in heavy resistance training conditions. There were 74 instances (60 in circuit training, and 14 in heavy resistance training) where a participant’s decrement exceeded a MD threshold, and 31 instances (8 in circuit training, 23 in heavy resistance training) of recorded improvements. Of all the firefighters, 72% had at least one instance of a decrement following circuit training, whereas 44% had at least one instance of an improvement following heavy resistance training. The participant that accumulated the most decrements (19 instances in total: circuit training = 16; heavy resistance training = 3) presented with a BMI and relative body fat ranked 2nd and 4th highest among the sample, respectively. During circuit training, peak heart rate was 101.5% (1st highest) and posttest blood lactate was 12.6 mmol/L (11th highest). During heavy resistance training, peak heart rate was 95.4% of age-predicted maximum (2nd highest) with a posttest blood lactate of 7.7 mmol/L (7th highest).

We identified correlates of decrements in neuromuscular outcomes. For circuit training, lesser baseline long jump distance was associated with greater peak force decrements (r = 0.635, *p* = 0.005). Greater post-exercise blood lactate and peak exercise heart rate (%MHR) were associated with greater long jump (r = −0.727, *p* < 0.001; r_sp_ = −0.701, *p* = 0.001) and lower body power decrements (r = −0.748, *p* < 0.001; r_sp_ = −0.666, *p* = 0.003).

### 3.2. Modified FBT-Validity and Reliability

The secondary aim evaluated the validity and reliability of the mFBT. For face validity, participants rated the relevance of the mFBT as follows: 33% indicated that it was “somewhat relevant” and 67% indicated it was “relevant” (Median (range) = 3.0 (2.0–3.0)). Regarding criterion related validity, the baseline mFBT PI (without cognitive load) was not correlated to baseline measures of single leg drop landing time-to-stabilization (r = 0.257, *p* = 0.303), or postural sway mean velocity (bipedal eyes open and closed, unipedal eyes open) (r = 0.175–0.453; *p =* 0.059–0.487). Table 6 and Table 7 and Figure 2 display the reliability metrics for the mFBT. Specifically, the intra-session, inter-trial reliability of the mFBT PI without and with cognitive load were ICC = 0.74 and 0.84, respectively. The inter-session reliability of the mFBT PI without and with cognitive load were ICC = 0.88 and 0.86, respectively.

## 4. Discussion

The purpose of this study was to investigate the impact of on-duty circuit and heavy resistance training on neuromuscular outcomes in firefighters. We hypothesized that circuit, but not heavy resistance training, would produce decrements in neuromuscular function. This hypothesis was partially supported. Indeed, both resistance training stimuli reduced muscular power immediately post exercise (Table 4), whereas circuit training also decreased isometric strength immediately post-exercise. These performance-based measures were taken to quantify the magnitude of fatigue induced by each exercise stimulus in an effort to provide context for the subsequent balance outcomes. It is important to note that we did not assess strength and power 10 min post-exercise and thus cannot speculate as to the status of these metrics at that time. With that said, these findings are noteworthy given that muscular strength and power are associated with firefighters’ occupational physical ability [34,35]. Although the present study did not evaluate the impact on subsequent occupational work output, Dennison et al. (2012) [5] reported that on-duty circuit training reduces occupational work rate by 10% 10 min post exercise. Interestingly, the relative decrements in strength and power reported immediately post exercise in the present study (7–15%) were similar to the reduction in work rate reported by Dennison et al. (2012) [5]. In a similar work, Mason and coworkers (2023) [4] indicated that high intensity functional training decreased firefighters’ work rate by 45% and increased air consumption by 27%, suggesting that greater resistance training intensity and volume, and reduced recovery periods negatively influence occupational performance. Future research is warranted to determine if biomotor decrements are associated with work output decrements, thus providing practical assessments to evaluate individual firefighter responses to various on-duty exercise stimuli. Furthermore, these findings align with Trivisonno and colleagues (2021) [7], who reported decrements in upper body peak force, countermovement jump height and velocity, and lower body rapid force 24–72 h after on-duty circuit training in firefighters. Also, the 24 h post exercise decrements in muscular strength and power (~3–10%) noted by Trivisonno et al. (2021) [7] were similar to the present study. Unlike Trivisonno et al. (2021) [7], the present study did not evaluate the time course of neuromuscular recovery at 10 min or longer.

Regarding balance outcomes, circuit training produced greater decrements than heavy resistance training in select bilateral postural sway measures with eyes-open and eyes-closed. However, despite non-significant trends (*p* = 0.066–0.098), there were no differences from baseline to post-exercise within the circuit training condition (Table 4), potentially due to the notable variability within the circuit training condition, reflecting heterogeneity in individual responses. To that point, an evaluation of individual responses of these select postural sway variables revealed that 17% of the firefighters were still experiencing impaired balance following circuit training (Table 5) and two of these firefighters chose not to perform the mFBT after circuit training due to fatigue. Indeed, the circuit training condition was more physically demanding than heavy resistance training. Specifically, circuit training resulted in greater physiological (post-blood lactate, average and peak heart rate) and perceptual (dyspnea and RPE) measures of strain. Research indicates that hyperventilation, metabolic products, and exercise intensities exceeding lactate threshold have been found to be related to a reduction in neuromuscular function [14]. Certainly, mean blood lactate following circuit training was elevated in the present study (13.0 ± 2.5 mmol/L). However, despite several non-significant trends, that was not the case in the present study.

Interestingly, there were no main or interaction effects for single-leg drop landing, single-leg stand, and mFBT measures. Certainly, these findings indicate the resistance training stimuli did not influence these unipedal and dynamic balance assessments. However, it is important to note that the reliability of the time-to-stabilization metric for the single-leg drop landing task was poor (ICC = 0.404) and may have influenced these results. In addition, the bilateral postural sway measurements and mFBT were conducted last in the testing sequence (i.e., ~15–20 min post-exercise) and thus firefighters may have recovered as research indicates that many balance metrics are restored 5–20 min post-stimulus [36,37,38,39,40,41,42,43,44,45]. However, the MD analysis indicated that 44% of firefighters improved at least one postural sway metric following the heavy resistance training bout (Table 5). Similarly, research indicates that balance may improve following exercise [46,47,48]. Improvements in neuromuscular function from pre-conditioning exercise may be related to post-activation potentiation [49], proactive adjustments due to prior knowledge of balance assessments [50], or simply being “warmed-up” in a non-fatiguing manner [48]. Interestingly, Paillard et al. [48] found that a warm-up improves postural control after about 10–15 min, which aligns with the recovery duration utilized in the present study and the subset of firefighters who experienced improvements in neuromuscular outcomes post-circuit and heavy resistance training. Although numerous additional studies have demonstrated balance decrements among firefighters, these decrements have typically been associated with firefighters performing occupational tasks or exercise in PPE during heat exposure [15,51,52,53]. This is important as heat stress can negatively impact functional balance [15,51,52]. The present study did not expose firefighters to heat or exercise in PPE.

The second hypothesis of this study was that fitness characteristics would be associated with decrements in neuromuscular function. Regarding circuit training, greater baseline long jump values were associated with lesser decrements in post-exercise isometric strength. Thus, it appears that possessing greater muscular power may provide tolerance to circuit training fatigue. Relevant research in firefighters has also demonstrated that greater baseline handgrip strength levels were inversely associated with the decrement in strength loss due to performing fatiguing occupational tasks [54]. In addition, greater markers of anaerobic and aerobic stress (blood lactate and heart rate) during the circuit training session were associated with greater post-exercise decrements in lower body power. These findings are intriguing and may have additional application in the fire service, as objective measures of internal load may be used to scale individualized exercise parameters to avoid undue fatigue during on-duty training.

The second aim of this study evaluated the validity and reliability of the mFBT with and without a cognitive load. The functional balance test has been utilized in numerous firefighter studies to evaluate dynamic balance following occupational tasks or exercise with or without PPE, as well as identifying associations with various outcomes of interest [15,51,52,55,56,57,58,59,60]. Regarding face validity, the majority of firefighters indicated that the mFBT was relevant to occupational movement patterns. However, regarding criterion validity, the mFBT PI was not related to force plate postural sway assessments, potentially due to the differences in the functional and static balance assessments utilized. Regarding inter-session reliability, the mFBT PI yielded acceptable coefficients with (ICC = 0.88) and without (ICC = 0.86) cognitive load integration, indicating stability over multiple days. The inter-trial reliability of the assessment also produced an acceptable coefficient (Table 6). Interestingly, there is a notable trend of improved PI over trials 1–6 (Figure 2, Table 6) as trials 4–6 were 10.6% faster, on average, than trials 1–3, suggesting that at least 3 familiarization trials are required. Anecdotally, firefighters appeared to utilize several trials to develop a comfortable and/or efficient movement strategy throughout the mFBT. These findings align with the inter-trial reliability study [25] of Punakallio’s original functional balance test (ICC = 0.78–0.96), which utilized 6 trials and identified a motor learning process necessitating at least 1 familiarization trial. Collectively, these findings are encouraging as they indicate that this inexpensive, user friendly, occupationally relevant balance assessment may have utility in the fire service. For instance, it may be used to evaluate the efficacy of balance training interventions, inform return-to-duty status, and be incorporated in annual physical ability and/or fitness evaluations as a marker of firefighter readiness. However, it is critical that additional research is conducted to determine if the PI is associated with injury incidence.

## 5. Limitations

There were several limitations in the present study. First, a BIA was utilized to estimate body fat percentage. Although BIA is a practical method, its accuracy may be affected by hydration status and should be interpreted with caution. In addition, this study only evaluated strength and power immediately post-exercise and balance 10 min post exercise, therefore caution is warranted when interpreting the results for neuromuscular outcomes after these time points. In addition, heat exposure was not evaluated in this study as there is a possibility that neuromuscular function and balance could be further impacted in warmer environments. Finally, this study was delimited to resistance training stimuli. Additional research is warranted to evaluate the impact of endurance training modalities and to evaluate the impact of training status on recovery outcomes.

## 6. Conclusions

Physically fit firefighters have a reduced risk of musculoskeletal injury and enhanced occupational readiness [61,62,63]. Thus, it is critical for firefighters to exercise regularly to optimize fitness. Research indicates that firefighters are less physically active off- compared to on-duty [64], as they likely experience many of the same barriers to exercise participation as the general population (e.g., lack of time and exercise resources, etc.) [65]. Thus, providing on-duty exercise time and resources facilitates a necessary behavior to enhance firefighter safety and readiness. In the event that an on-duty circuit training session is interrupted with an emergency response, trained firefighters still tend to perform occupational tasks more efficiently than non-fatigued untrained firefighters [5]. Therefore, tactical strength and conditioning practitioners and fire department leadership should consider cost-effective and feasible countermeasures to safely enhance firefighter fitness. For instance, it is cautionary to suggest that heavy resistance, and especially circuit training, be performed during low-volume call times or near the end of a shift to reduce the likelihood of responding to an emergency in a fatigued state. However, despite the occupational specificity of circuit training, it may be more safely performed off-duty due to the use of high volume and minimal recovery training parameters that induce high levels of internal strain and marked post-exercise decrements in occupational readiness [4]. Regardless of the training method, firefighters and tactical strength and conditioning practitioners should utilize the progression principle to systematically introduce novel exercise stimuli and evaluate individual recovery responses to modify training parameters accordingly. Regular participation in a scaled, periodized program will produce an improved fitness profile that may reduce post-resistance training neuromuscular decrements and enhance occupational readiness.

Firefighters in the present study experienced decrements in muscular strength and/or power immediately following exercise but most did not experience decrements in balance 10 min post resistance training. Possessing greater levels of muscular strength and power may attenuate decrements in these attributes following on-duty circuit training sessions. Finally, the mFBT was found to have adequate face validity, acceptable test–retest reliability, and may have utility in the fire service.

## Figures and Tables

**Figure 1 healthcare-13-01278-f001:**
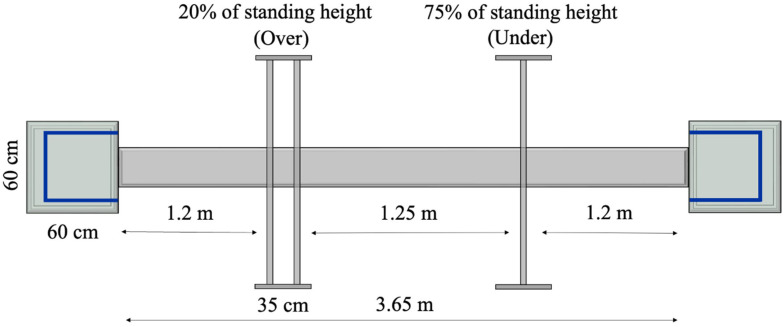
Schematic of the modified functional balance test.

**Figure 2 healthcare-13-01278-f002:**
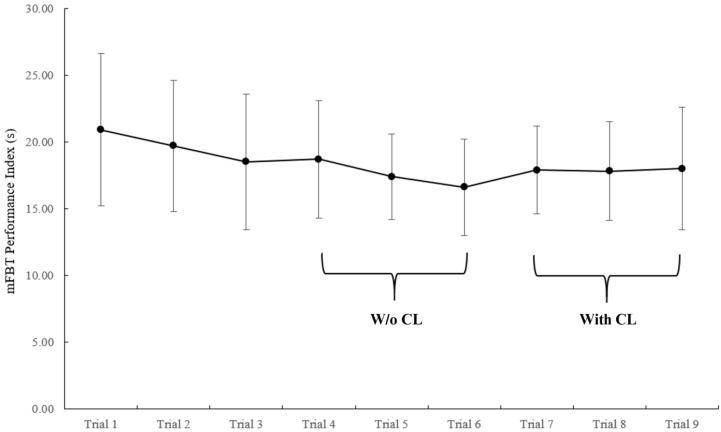
Inter-trial modified functional balance test (mFBT) performance index with and without integration of the cognitive load task in 18 firefighters during the familiarization session. The mFBT PI stabilizes after 3 trials in naïve performers. Values represent mean ± standard deviation. W/o CL: without cognitive load; CL: cognitive load.

**Table 1 healthcare-13-01278-t001:** Summary of the testing session composition and protocol sequence.

Session 1	Session 2	Session 3
Anthropometrics SLDL (PPE) Postural Sway (PPE) Functional Balance (PPE) Warm-up IMTP Long Jump	**Pre-exercise** SLDL (PPE) Postural Sway (PPE) Functional Balance (PPE) Blood Lactate RPE/Dyspnea/Thermal Skin Temperature Warm-up IMTP Long Jump	**Pre-exercise** SLDL (PPE) Postural Sway (PPE) Functional Balance (PPE) Blood Lactate RPE/Dyspnea/Thermal Skin Temperature Warm-up IMTP Long Jump
5 RM Strength Assessment	**Exercise:** HRT or CT Condition	**Exercise:** HRT or CT Condition
	**Recovery**	**Recovery**
	IMTP (0 min-post)	IMTP (0 min-post)
	Long Jump (0 min-post)	Long Jump (0 min-post)
	Skin Temperature (0 min-post)	Skin Temperature (0 min-post)
	RPE/Dyspnea/Thermal (0 min-post)	RPE/Dyspnea/Thermal (0 min-post)
	Blood Lactate (5 min-post)	Blood Lactate (5 min-post)
	**10 min post-exercise**	**10 min post-exercise**
	SLDL (PPE)	SLDL (PPE)
	Postural Sway (PPE)	Postural Sway (PPE)
	Functional Balance (PPE)	Functional Balance (PPE)

HRT: heavy resistance training; CT: circuit training; PPE: performed while wearing personal protective equipment; SLDL: single-leg drop landing; IMTP: isometric midthigh pull; RPE: rating of perceived exertion; RM: repetition maximum.

**Table 2 healthcare-13-01278-t002:** Descriptive comparison of the Swedish occupational fatigue inventory outcomes prior to circuit (CT) and heavy resistance training (HRT) sessions in 16 firefighters.

	CT	HRT
Lack of energy	0.88 ± 0.90	0.80 ± 0.87
Lack of motivation	0.72 ± 1.13	0.54 ± 1.21
Sleepiness	0.82 ± 0.81	0.66 ± 0.49
Physical exertion	0.24 ± 0.42	0.22 ± 0.48
Physical discomfort	0.88 ± 0.72	0.83 ± 0.78

Values reflect mean ± standard deviation. Two firefighters did not complete the entire SOFI survey; therefore, their data were omitted (n = 16). There were no differences in constructs across conditions (*p* ≥ 0.158).

**Table 3 healthcare-13-01278-t003:** Comparison of physiological and perceptual metrics of circuit training (CT) and heavy resistance training (HRT) conditions in 18 firefighters.

	CT	HRT
Blood lactate_pre_ (mmol/L)	2.0 ± 0.9 ^‡^	2.1 ± 1.2 ^‡^
Blood lactate_post_ (mmol/L) *	13.0 ± 2.5 ^‡^	6.6 ± 2.2 ^‡^
HR_peak_ (%MHR) *	94.4 ± 3.9	88.0 ± 7.3
HR_average_ (%MHR) *	81.5 ± 4.3	69.1 ± 7.0
Thermal Strain_pre_	0.4 ± 1.0	0.5 ± 0.9
Thermal Strain_post_	1.8 ± 0.7	1.4 ± 0.8
Dyspnea_pre_	0.8 ± 0.7 ^‡^	1.0 ± 0.9 ^‡^
Dyspnea_post_ *	3.9 ± 2.0 ^‡^	2.6 ± 1.1 ^‡^
RPE_pre_	1.6 ± 0.8 ^‡^	1.6 ± 0.9 ^‡^
RPE_post_ *	6.3 ± 2.1 ^‡^	4.7 ± 1.4 ^‡^
Temperature_pre_ (°C)	36.2 ± 0.4	36.1 ± 0.3
Temperature_post_ (°C)	35.8 ± 1.2	34.7 ± 2.5
Urine specific gravity (USG) ^†^	1.017 ± 0.009	1.032 ± 0.070

Values reflect mean ± standard deviation. * Significant difference between conditions (*p* < 0.05); ^‡^ significant difference within conditions (*p* < 0.05); ^†^ euhydration ≤ 1.020; %MHR = Percent of maximal age-predicted heart rate; RPE: rating of perceived exertion; heart rate values are reported in n = 17(HR_peak_) and n = 13 (HR_average_) participants due to technological issues. Circuit training produced greater blood lactate (*p* < 0.001), peak (*p* = 0.002) and average heart rate (*p* < 0.001), dyspnea (*p* = 0.005), and RPE (*p* = 0.007) compared to heavy resistance training.

**Table 4 healthcare-13-01278-t004:** Comparison of circuit training (CT) versus heavy resistance training (HRT) neuromuscular outcomes in 18 structural firefighters.

	CT Rel Diff (%)	HRT Rel Diff (%)	*p*-Value ^‡^	Partial Eta Squared ^‡^	Power ^‡^
Strength and Power Performance					
Peak Force *^,†^	−6.5 ± 5.1	−2.7 ± 6.2	0.009	0.340	0.796
Long Jump *^,†^	−7.4 ± 7.7	−3.1 ± 2.5	0.017	0.294	0.708
Lower Body Power *^,†^	−14.8 ± 15.0	−6.7 ± 5.6	0.017	0.294	0.708
Single Leg Drop Land					
Time to Stabilization	5.1 ± 43.4	7.6 ± 32.7	0.806	0.004	0.056
Peak Force (n = 17)	−2.7 ± 10.5	3.8 ± 11.5	0.148	0.126	0.298
Postural Sway (Eyes Open)					
Mean Velocity *	13.6 ± 30.6	0.2 ± 19.8	0.043	0.219	0.539
Mean Velocity-Anterior–Posterior *	13.0 ± 29.9	0.3 ± 20.8	0.049	0.208	0.514
Mean Velocity-Medial–Lateral	16.3 ± 37.3	0.7 ± 17.7	0.088	0.162	0.401
Excursion *	13.5 ± 30.5	0.2 ± 19.7	0.046	0.214	0.529
COP Range-Anterior–Posterior	18.5 ± 37.9	19.9 ± 39.1	0.485	0.029	0.103
COP Range-Medial–Lateral	42.0 ± 91.6	14.3 ± 54.1	0.19	0.099	0.251
Postural Sway (Eyes Closed)					
Mean Velocity	11.1 ± 24.4	−3.2 ± 22.9	0.086	0.163	0.405
Mean Velocity-Anterior–Posterior	11.6 ± 25.3	−2.5 ± 23.7	0.098	0.153	0.379
Mean Velocity-Medial–Lateral *	10.9 ± 24.1	−10.4 ± 18.9	0.032	0.243	0.595
Excursion	11.1 ± 24.4	−3.2 ± 23.0	0.087	0.162	0.402
COP Range-Anterior–Posterior	26.2 ± 41.1	7.0 ± 41.0	0.423	0.038	0.121
COP Range-Medial–Lateral *	35.6 ± 59.6	−6.3 ± 34.5	0.011	0.322	0.763
Single-Leg Stand					
Mean Velocity	3.9 ± 25.3	−6.3 ± 16.7	0.117	0.138	0.344
Mean Velocity-Anterior–Posterior	6.8 ± 24.7	−4.7 ± 13.6	0.054	0.201	0.496
Mean Velocity-Medial–Lateral	2.2 ± 26.3	−7.4 ± 20.2	0.182	0.102	0.260
Excursion	3.9 ± 25.4	−6.2 ± 16.7	0.119	0.137	0.341
COP Range-Anterior–Posterior	28.3 ± 51.5	6.9 ± 27.1	0.172	0.107	0.270
COP Range-Medial–Lateral	6.8 ± 33.6	−2.1 ± 24.1	0.119	0.137	0.341
Modified Functional Balance Test (n = 16)					
mFBT Time	6.0 ± 13.4	2.6 ± 7.9	0.344	0.060	0.150
mFBT Performance Index	5.2 ± 18.2	3.0 ± 14.2	0.585	0.020	0.082
mFBT Time (w/Cognitive Load)	3.9 ± 11.6	3.1 ± 7.1	0.820	0.004	0.055
mFBT Performance Index (w/Cognitive Load)	7.6 ± 19.6	4.7 ± 11.4	0.587	0.020	0.081

Values reflect mean ± standard deviation. Rel Diff: Relative difference. Rel Diff = ((post-exercise value − pre-exercise value)/pre-exercise value) × 100. ^‡^
*p*-value, partial eta squared and power describe condition × time interaction effects; * significant condition × time effect (*p* < 0.05); ^†^ significant main effect of time (*p* < 0.05); COP: center of pressure; mFBT: modified functional balance test. Circuit training decreased peak force, long jump distance and lower body power (*p* ≤ 0.001); whereas heavy resistance training decreased long jump distance and lower body power (*p* < 0.001), but not peak force (*p* = 0.102). Despite interaction effects on several postural sway metrics, post hoc analysis indicated that within condition comparisons were not different than baseline.

**Table 5 healthcare-13-01278-t005:** Total and relative number of participants outside of the minimal difference (MD) threshold for circuit training (CT) and heavy resistance training (HRT) conditions in 18 firefighters.

	CT		HRT	
	Decrement	Improvement	Decrement	Improvement
Variable (n = 18)	% Outside MD (Frequency)		% Outside MD (Frequency)	
Peak Force	22.0 (4)	---	5.6 (1)	5.6 (1)
Long Jump	50.0 (9)	---	---	---
Lower Body Power	50.0 (9)	---	---	---
Single Leg Drop Landing-TTS	---	---	---	11.1 (2)
Postural Sway-EO-Mean Velocity	16.7 (3)	---	5.6 (1)	5.6 (1)
Postural Sway-EO-APMV	16.7 (3)	---	5.6 (1)	5.6 (1)
Postural Sway-EO-MLMV	22.2 (4)	---	---	5.6 (1)
Postural Sway-EO-Excursion	16.7 (3)	---	5.6 (1)	5.6 (1)
Postural Sway-EO-COPAP	11.1 (2)	5.6 (1)	11.1 (2)	---
Postural Sway-EO-COPML	16.7 (3)	---	16.7 (3)	---
Postural Sway-EC-Mean Velocity	---	---	---	11.1 (2)
Postural Sway-EC-APMV	---	---	---	11.1 (2)
Postural Sway-EC-MLMV	---	---	---	5.6 (1)
Postural Sway-EC-Excursion	---	---	---	11.1 (2)
Postural Sway-EC-COPAP	16.7 (3)	---	11.1 (2)	---
Postural Sway-EC-COPML	5.6 (1)	---	---	---
Single Leg Stand-MV	5.6 (1)	5.6 (1)	---	5.6 (1)
Single Leg Stand-APMV	5.6 (1)	---	---	5.6 (1)
Single Leg Stand-MLMV	5.6 (1)	5.6 (1)	---	11.1 (2)
Single Leg Stand-Excursion	5.6 (1)	5.6 (1)	---	5.6 (1)
Single Leg Stand-COPAP	11.1 (2)	---	---	---
Single Leg Stand-COPML	5.6 (1)	---	5.6 (1)	5.6 (1)
Variable (n = 16)				
mFBT Time	18.8 (3)	---	---	6.3 (1)
mFBT Performance Index	12.5 (2)	6.3 (1)	6.3 (1)	6.3 (1)
mFBT Time (Cognitive Load)	12.5 (2)	6.3 (1)	---	6.3 (1)
mFBT Performance Index (Cognitive Load)	12.5 (2)	12.5 (2)	6.3 (1)	---

IMTP: isometric midthigh pull; TTS: time to stabilization; EO: eyes open; EC: eyes closed; APMV: anterior–posterior mean velocity; MLMV: medial lateral mean velocity; COPAP: center of pressure anterior–posterior; COPML: center of pressure medial lateral; mFBT: modified functional balance test. Despite individual variability in responses to the resistance training stimuli, circuit training tended to negatively impact neuromuscular outcomes more than heavy resistance training immediately post-exercise.

**Table 6 healthcare-13-01278-t006:** Inter-trial reliability of the modified functional balance test with and without integration of the cognitive load task in 18 firefighters during the familiarization session.

	W/o CL Trials #1–3	ICC (95% CI)	W/o CL Trials #4–6	ICC (95% CI)	With CL Trials #7–9	ICC (95% CI)
mFBT Time (s)	17.5 ± 4.8	0.843 (0.680–0.934)	15.4 ± 3.1	0.875 (0.659–0.954)	15.5 ± 3.2	0.880 (0.759–0.949)
mFBT Performance Index (s)	19.7 ± 5.0	0.826 (0.634–0.928)	17.6 ± 3.5	0.739 (0.493–0.886)	17.9 ± 3.7	0.838 (0.685–0.930)

W/o: without; CL: cognitive load; mFBT: modified functional balance test. ICC: Intraclass correlation coefficient. The mFBT PI yielded acceptable inter-trial reliability with and without a cognitive load.

**Table 7 healthcare-13-01278-t007:** Inter-session (pre-exercise) reliability of the modified functional balance test with and without integration of the cognitive load task in 18 firefighters.

	Session 2	Session 3	ICC (95% CI)	SEM	MD	CoV (%)
mFBT Time (s)	13.7 ± 2.9	13.6 ± 2.9	0.915 (0.788–0.967)	0.88	2.44	6.43
mFBT Performance Index (s)	15.1 ± 3.2	15.1 ± 3.2	0.884 (0.716–0.955)	1.11	3.09	7.38
mFBT Time (Cognitive Load) (s)	13.8 ± 2.8	13.8 ± 2.7	0.946 (0.861–0.979)	0.66	1.83	4.80
mFBT Performance Index (Cognitive Load) (s)	15.1 ± 3.1	15.3 ± 2.8	0.863 (0.674–0.947)	1.12	3.10	7.32

mFBT: modified functional balance test. ICC: intraclass correlation coefficient; SEM: standard error of measurement; MD: minimal difference; CoV: coefficient of variation. The mFBT PI yielded acceptable inter-session reliability with and without a cognitive load.

## Data Availability

The datasets presented in this article are not readily available due to privacy concerns. Requests to access the datasets should be directed to the corresponding authors.

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
