# Peer review of "Effect of On-Duty Resistance Training Fatigue on Neuromuscular Function in Structural Firefighters"

_healthcare, 2025, doi:10.3390/healthcare13111278_

Round 1
Reviewer 1 Report
Comments and Suggestions for Authors
Dear author, I have read your study and found some deficiencies. You can find these deficiencies in the PDF file below. In addition, when it comes to the study, the study is original and at a level to contribute to the literature. It is presented meticulously and every detail is explained.

Author Response
Thank you very much for taking the time to review this manuscript. Please see the attachment for responses to feedback, and please see the tracked changes in the manuscript to see the specific edits made.

Reviewer 2 Report
Comments and Suggestions for Authors
The manuscript is well written and addresses an important practical issue in occupational training for firefighters. Below are some detailed comments and suggestions to further strengthen your work:
1. Introduction:
-
The background is solid, providing sufficient context on the injury risks and fitness requirements for firefighters.
-
Suggestion: Consider elaborating on the physiological mechanisms by which circuit training may produce greater neuromuscular fatigue than heavy resistance training. A short discussion of cumulative fatigue due to higher cardiovascular and metabolic load in circuit training could help frame your hypothesis more strongly and align it with current physiological theory.
2. Methods:
-
The methodology section is detailed and replicable. The design is appropriate, with a crossover approach that enhances internal validity.
-
Suggestion: Although selecting performance and balance assessments is appropriate, it would improve the clarity of your rationale to include a brief explanation or reference justifying why those specific neuromuscular tests and balance metrics were chosen. For instance, do they best mimic firefighting tasks or represent common performance indicators in occupational health research?
-
Suggestion: Address whether any familiarization sessions were conducted for the balance tests to minimize learning effects between the first and second sessions. If none were done, acknowledge this as a limitation, as balance can improve with repeated exposure regardless of intervention.
3. Results:
-
The results are presented using appropriate statistical analysis.
-
Suggestion: While tables provide all necessary information, consider adding visual representations (e.g., bar graphs with error bars for significant differences, line plots for time-based comparisons) to make the key findings more accessible and visually interpretable, especially for practitioner audiences.
-
Suggestion: Highlight key results in table captions (e.g., “SJ height significantly decreased after CIR vs. RES, p < 0.05”) to make navigation easier.
4. Discussion:
-
The discussion effectively interprets the findings and situates them within the existing literature.
-
Suggestion: Expand the applied implications of the findings. For example, how might these acute effects inform the timing and sequencing of physical tasks within a firefighter’s daily schedule? Should circuit training be avoided before tasks requiring high neuromuscular control? Practical guidelines based on your results could increase the real-world impact of your study.
-
Suggestion: Consider highlighting the variability in individual responses to training modalities and discuss the potential for individualized or periodized training plans in firefighter populations. This would align with current trends in occupational exercise prescription.
5. Language and Style:
-
The English language is fluent and professional throughout. Sentence structure, vocabulary, and grammar are all appropriate for academic publication. No revisions are needed.
6. Conclusion:
-
The conclusion is valid and well-aligned with the data.
-
Suggestion: Reinforce the need for context-aware training designs, emphasizing how different resistance modalities impact performance recovery. You could also briefly suggest directions for future research, such as longer-term training interventions or inclusion of more diverse firefighting populations.
Author Response

(The authors gave the same response as above.)
